# Evaluating task shifting to the clinical technologist in Dutch healthcare: A mixed methods study

**Maarten de Haan** [1,2,3]*, **Yvonne van Eijk-Hustings** [1,2], **Monique Bessems-Beks** [1,2], **Daisy De Bruijn-Geraets** [1,2], **Carmen Dirksen** [1,2,4], **Hubertus Vrijhoef** [1,5]

1 Department of Clinical Epidemiology and Medical Technology Assessment (KEMTA), Maastricht University Medical Center, Maastricht, The Netherlands, 2 School for Public Health and Primary Care (CAPHRI), Maastricht University, Maastricht, The Netherlands, 3 National Healthcare Institute (Zorginstituut Nederland), Diemen, The Netherlands, 4 Clinical Trial Center Maastricht (CTCM), Maastricht, The Netherlands, 5 Panaxea b.v., Amsterdam, The Netherlands

* mah.de.haan@mumc.nl

## Abstract

### Background

Expected rise in the demand for healthcare and a dropping supply of healthcare professionals, has generated an increased interest in the most effective deployment of healthcare professionals. Consequently, task shifting has become a common strategy to redistribute tasks between established professional groups, however, little is known about the effects of shifting tasks to emerging professional groups. The aim of this study was to evaluate a legal amendment to facilitate task shifting to an emerging profession in Dutch healthcare: Clinical Technologists (CTs). CTs were introduced and provided an Extended Scope of Practice (ESP) to perform nine 'reserved procedures' independently.

### Methods

A concurrent multi-phase mixed methods study was used to evaluate whether a legal amendment to facilitate task shifting to CTs was effective and efficient.

### Results

The results show that CTs use their ESP frequently to perform five categories of reserved procedures independently and suggest that the ESP increased the efficiency of care delivery for those procedures. Additionally, the findings highlight that task shifting was influenced by the setting in which CTs worked, time allotted to patient-contact as well as external factors (e.g., financing).

### Conclusions

This study provides tentative lessons for policymakers on how task shifting to emerging professional groups can be improved. Providing a legal amendment to facilitate task shifting to CTs seems to be effective and efficient. However, it also poses multiple challenges. While

**Data Availability Statement:** We have asked Prof. Dr. Manuela Joore to serve as independent reviewer for assessing requests for access to our data. She is Professor of Health Technology

Assessment & Decision Making at Maastricht University and Director of the Department of Clinical Epidemiology and Medical Technology Assessment (KEMTA) at Maastricht University Medical Center. She was not involved in conducting the study but has direct access to the data. Data is not publicly shared because despite our best efforts to anonymize the data-set, individuals are still likely to be identified due to the small respondent group and specific expertise of individual Clinical Technologists. A request to see the data underlying the results presented in the study can be made through Prof. Dr. Manuele Joore (m.joore@mumc.nl).

**Funding:** This work was supported by the Dutch Ministry of Health, Wellbeing and Sports [Ministerie van Volksgezondheid, Welzijn en Sport] Grant number: 322333. The funder had no role in study design, data collection and analysis, decision to publish, or preparation of the manuscript.

**Competing interests:** The authors have declared that no competing interests exist.

established professional groups can face similar challenges, it is likely that these are exacerbated for emerging professional groups, particularly when shifting tasks occurs horizontally.

## 1. Introduction

Task shifting, the transfer of work and responsibilities from one professional group to another [1], is a common strategy to redistribute tasks between established professional groups. Usually this redistribution takes place from higher schooled professional groups to lesser schooled professional groups. In developed countries, task shifting has been predominantly used to improve quality of care and limit healthcare costs [2, 3]. Task shifting is increasing, but research is scarce [2]. Studies evaluating task shifting to established professionals groups, like Nurse Practitioners (NP's) and Physician Assistants (PA's) have reported higher quality of care, higher patient satisfaction and improvements in clinical outcomes [3] and often saved time [4]. The evidence for this is strongest with nurses [5]. Consequently, even though responsibilities and performed procedures differs greatly between and within professional groups, task shifting to such professional groups is widely considered of added value [6]. However, the relationship between structure, process and outcomes measures are often not included in such studies [7] limiting their generalizability to emerging professional groups. Due to the rise of new healthcare technologies, changing patient preferences, new care models and educational opportunities, it is likely that new professional roles and groups with different skills will be needed to ensure healthcare systems can continue to meet the highest standard of care [8, 9].

The Netherlands is one of the few countries experimenting with task shifting to both established and emerging professional groups. One way in which the Dutch Ministry of Health, Wellbeing and Sports (HWS) has aimed to facilitate task shifting, is by providing a temporary Expanded Scope of Practice (ESP) to professional groups for five years, during which the ESP is evaluated to determine if it should be made permanent. Such an experiment has already led to a permanent ESP for established professional groups such as NP's and PA's. However, between 2014–2018, they also experimented with a new professional group, the Clinical Technologist (CT). CTs have completed a six-year university curriculum, which combines traditional medical courses with engineering, physics and math [10] and are geared towards using, improving and creating complex medical technologies [11]. CT education consists of a three year's Bachelor and three year's Master program, during which CTs complete as least two year's clinical internships in which they combine patient care and research. Technical Medical Skills are an important part of the curriculum and CTs are trained to perform (reserved) procedures, demonstrate appropriate professional behavior, laboratory skills, communication skills and diagnostic and therapeutic skills (including physical examinations). Students can choose between five specializations: robotica & imaging, medical signaling, reconstructive medicine, medical imaging & interventions or medical sensing & stimulation. The first CTs graduated in 2009 and currently more than 500 CTs have graduated [12].

Providing an ESP is a legal necessity to facilitate task shifting in the Netherlands, since the Individual Healthcare Professions Act (IHCP-Act) restricts the performance of 14 categories of medical procedures, known as 'reserved procedures' [13]. Reserved procedures are deemed too dangerous to be performed independently by unqualified or uncertified groups and may thus only be indicated, performed and delegated by professionals with an independent authority (e.g., Medical Doctor) or performed by professionals with a functional independent authority (e.g., nurses). The latter group is only allowed to perform such procedures after an order from a professional that has an independent authority. All other professionals are only allowed

to perform reserved procedures after an order and under supervision from a professional with an independent authority [14, 15].

The legal amendment providing a temporary ESP for CTs included an independent authority to perform nine categories of reserved procedures, namely: cardioversion, catheterizations, defibrillation, endoscopy, injections, lithotripsy, procedures with (ionizing) radiation, punctures and surgical procedures. Commissioned by the Dutch Ministry of HWS, we evaluated the effectiveness and efficiency of the temporary ESP to aid the Ministry's decision-making process on whether or not to make the ESP permanent. Consequently, in the context of this study 'effectiveness' and 'efficiency' were defined in relation to the legal amendment created by the ESP and the focus of CTs on using, improving and creating complex medical technologies. As such, for the legal amendment to be 'effective' it should be used by CTs to perform reserved procedures independently, and to be 'efficient' this should lead to these procedures taking up less time than when those procedures are first delegated to CTs by an MD or another professional with an independent authority to perform reserved procedures [16].

## 2. Methods

Quantitative (QUAN) and Qualitative (QUAL) data were collected in a concurrent multiphase mixed methods study (MMR) [17–19] conducted between September 2014 and October 2017. A mixed methods approach was chosen because the research questions asked were broad and called for an evaluation on multiple perspectives that could not be answered by either QUAN or QUAL data alone. Furthermore, using both QUAN and QUAL data improves the credibility and confirmability of the overall results [20].

The design and theoretical framework used, were derived from a previous evaluation of the temporary ESP of the NP/PA in the Netherlands [21]. QUAN and QUAL data were considered of equal importance and triangulated to collectively answer the research questions. During the study, the research team was supported by an advisory group consisting of important stakeholders (e.g., professional bodies, educational institutions and experts). The advisory group helped identify and recruit stakeholders for interviews and commented on preliminary research results. Fig 1 provides an overview of our study design and response.

### 2.1 Conceptual framework

To evaluate the legal amendment providing an ESP for CTs, the *Conceptual Framework for Evaluating the Nurse Practitioner Role in Acute Care Settings* [22] by Sidani & Irvine, was translated from nurse to CT. The framework is based on a broader model for evaluating quality of care, introduced by Donabedian [23], and differentiates between structure-, process- and outcome variables. An important proposition of this model is that structure variables have an effect on both process variables and outcome variables, as well as that process variables have an effect on outcome variables.

### 2.2 Sampling

The study population consisted of–at the time of this study–all graduated CTs within the Netherlands, a subpopulation of patients whom had undergone a reserved procedure performed by a CT, and Medical Doctors (MDs) whom worked intensely with CTs and regularly delegated reserved procedures to CTs. CTs, patients and MDs were asked to participate in both the QUAN- and QUAL data collection. In addition, stakeholders involved in the deployment of CTs or their ability to perform reserved procedures (e.g. hospital management, policy advisors, other healthcare professionals, educators, interest groups) were asked to participate in the QUAL data collection.

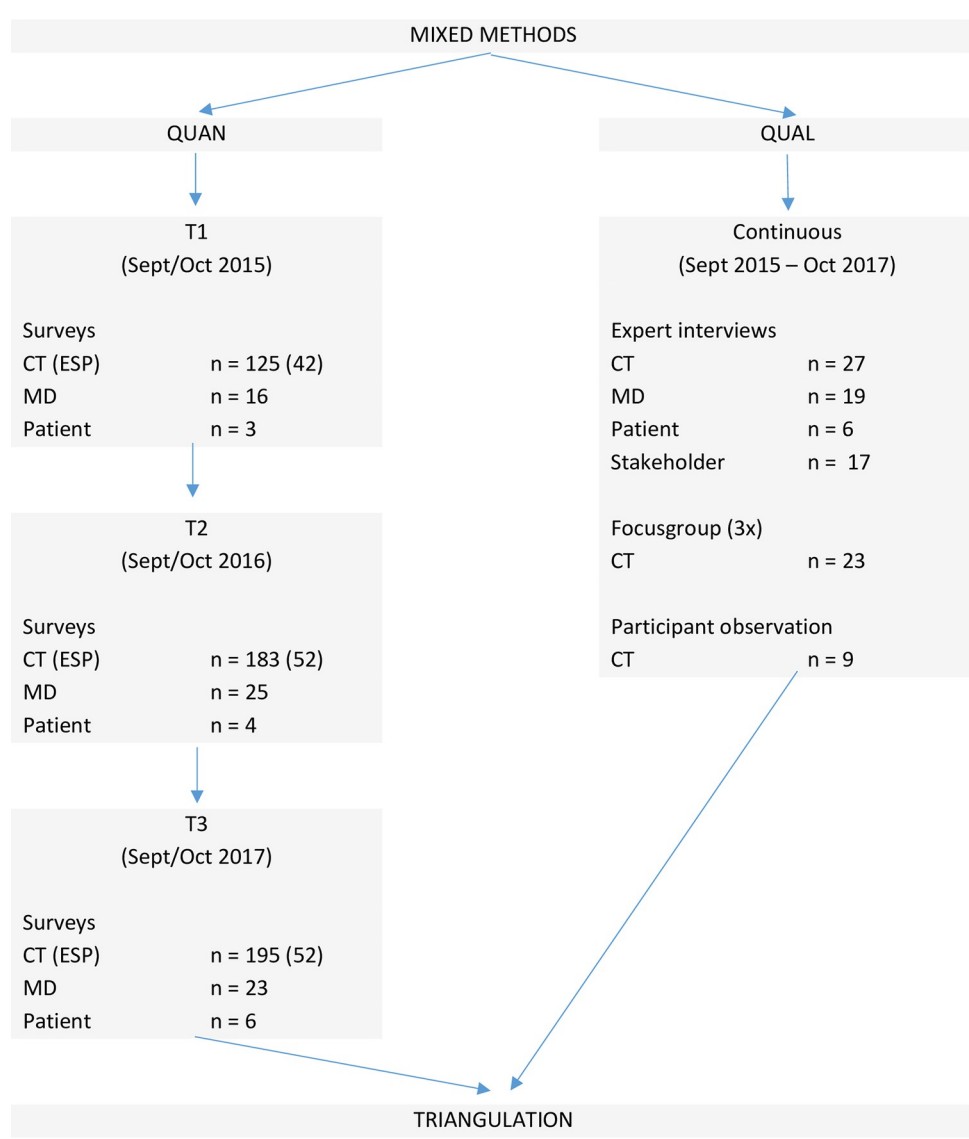

**Figure legend**
*CT = Clinical Technologist*
*MD = Medical Doctor*
*QUAL = Qualitative*
*QUAN = Quantitative*
*T = measurement point*

**Fig 1. Study design & response.**

## 2.3 Quantitative data collection

QUAN data were collected by means of digital surveys at three measurement points, from a purposive sample of CT graduates. CTs that did not perform reserved procedures were asked to fill in a short survey while CTs that did perform reserved procedures were asked to fill in an extended survey. Per measurement point, CTs that performed reserved procedures were asked to approach two MDs and (a maximum of) five patients to also fill out surveys.

Data collection was guided by the Sidani & Irvine [22] framework (see Table 1 for the operationalization) and consisted of i) structure, ii) process and iii) outcome variables.

Structure variables were collected for each group (CTs, MDs and patients) and included background characteristics such as: sex, age, master specialization (for CT), discipline, job title (for CTs), years of experience and organizational features (e.g., hospital type). Patients were, in addition to sex and age, asked about their education, disease type, ethnicity, family structure, general health and co-morbidities.

Process variables were collected from CTs and MDs and included the number of intake- and control consultations performed per week and the time spent per week on patient- and non-patient related activities. Patient related activities included: intake- and control consultations, medical procedures, patient rounds, patient related deliberations, organization-bound deliberations, multidisciplinary-deliberations and 'other'. Non patient related activities included: administration, schooling, scientific research, quality-improvement, teaching and 'other'.

**Table 1. Operationalization of Sidani & Irvine framework for this study.**

| Structure variables | |
|---|---|
| QUAN | QUAL |
| **CT/MD background characteristics** <br> *Sex, age, master specialization (for CT), discipline, job title (for CTs), years of experience* | **CT/MD background characteristics** <br> *Job title, master specialization (for CT), medical discipline, years of experience* |
| **Patient background characteristics** <br> *Sex, age, education, disease type, ethnicity, family structure, general health, co-morbidities* | **Patient background characteristics** <br> *Disease type, general health, co-morbidities* |
| **Organizational features** <br> *Organization type* | **Organizational features** <br> *Organization type, estimated number of CTs in workforce, policy on hiring CTs* |
| **Process variables** | |
| QUAN | QUAL |
| **Intake and control consultations (CTs/MDs)** <br> *Mean number of individual intake- and control-consultations per week* | **Intake and control consultations (CTs)** <br> *Role of CT in intake- and control consultations* |
| **Weekly activities (CTs/MDs)** <br> *Mean time spent on patient- and non-patient bound activities per week* | **Weekly activities (CTs)** <br> *Role of CT in patient- and non-patient bound activities.* |
| **Outcome variables** | |
| QUAN | QUAL |
| **Effectiveness** <br> *Mean number of reserved procedures indicated, performed and delegated by CTs (per month) and level of independence of CTs* | **Effectiveness** <br> *Indication, performance and delegation of reserved procedures, level of independence, method of testing competence, views on reserved procedures by CTs, MDs and stakeholders* |
| **Efficiency** <br> *Duration for indicating, performing and delegating reserved procedures of CTs and duration of delegating reserved procedures from MD to CT.* | **Efficiency** <br> *Estimated difference (in time and possible other resources) in duration between independent and non-independent indication, performance and delegation of reserved procedure by CT.* |

Legend

CT = Clinical Technologist

MD = Medical Doctor

QUAL = qualitative

QUAN = quantitative

Outcome variables were collected from CTs and MDs and focused on the extent to which CTs used the ESP (i.e. number of reserved procedures indicated, performed and delegated per month, and level of independence of CT) and what effects this has on the efficiency of care (i.e. duration for indicating, performing and delegating reserved procedures of CTs, duration for delegating reserved procedure from MD to CT).

## 2.4 Qualitative data collection

Multiple forms of QUAL data collection were used to increase the trustworthiness of the results. QUAL data were collected continuously and consisted of expert interviews (with CTs, MDs, Patients and Stakeholders), focus groups (CTs) and observations (CTs). CTs and MDs were approached by means of '(stratified) purposive sampling' [24] informed by QUAN data collection. The following strata were used in our sampling strategy: use of ESP, institution type, (official) job title and (medical) discipline. CTs were asked to approach MDs and patients willing to participate.

Individual interviews and focus group interviews had a semi-structured set-up through the use of protocols [25]. Observations took place by means of 'participant observation' [26]. The duration of individual interviews was typically 45 minutes, focus groups lasted for 2 hours and the duration of observations varied between 2–6 hours. MdH, YvEH, MBB and HV were involved in the QUAL data collection and all had prior training and experience in conducting QUAL research.

## 2.5 Data analysis

QUAN and QUAL data were first analyzed separately prior to being combined through the triangulation process in order to collectively answer the research question(s).

**2.5.1 Quantitative data analysis.** Initially it was our intent to follow the group of CTs performing reserved procedures throughout the experimentation period and conduct longitudinal analysis. However, this was not possible due to the small group of CTs performing reserved procedures during all three measurement points. Consequently, statistical testing was limited to several key structure variables. For all other variables, descriptive statistics (e.g., numbers and percentages) are provided per measurement point. An overview of the original plan is available in Dutch [27].

For each measurement point structure variables were analyzed per group (CT, MD, patient) and subgroup (e.g., no ESP versus ESP). Also, differences between the three measurement points were analyzed for key background variables (age and sex). Normality was inspected visually through histograms and by means of the Kolmogorov-Smirnov test. For continuous variables (e.g., age) means and standard deviations (SDs) were calculated for normally distributed data, and for non-normally distributed data medians and inter quartile ranges (IQRs) were calculated. For categorical variables (e.g., medical discipline) frequencies and percentages were calculated. To test for differences between groups (e.g., no ESP versus ESP) per measurement point an independent sample t-test was used for continuous variables (or Mann-Whitney for non-normally distributed data) and a Pearson's chi-squared test (with one degree of freedom) was used for categorical variables. For categorical data whereby one (or several) categories were reported by less than five respondents, a Fisher's exact test was also done. To test for differences between measurement points a Kruskal Wallis test was used for continuous variables and a Pearson's chi-squared test was used for categorical variables.

At each measurement point, monthly performance of reserved procedures was determined per category (e.g., catheterization) by adding all subcategories (e.g. bladder-catheterization and heart-catheterization) together per individual participant, subsequently a group-mean was

calculated. The duration was divided in six categories ('1–10 minutes', '11–20 minutes', '21–30 minutes', '31–40 minutes', '41–50 minutes' and 'more than 50 minutes'). Means of the frequencies and categories of duration were only calculated if a reserved procedure category (e.g., catheterization) was performed by more than five participants during each measurement point. CTs could also rank the level of independence of the performed procedures into seven categories, namely:

1. Indication MD, stipulated by protocol;

2. Indication MD, written order;

3. Indication MD, verbal order;

4. Indication CT, after consulting MD;

5. Indication CT, no consultation;

6. Indication CT, delegation to other professional; and

7. Indication different.

A similar scale, ranging from 1–5, was used to determine the level of independence of CTs (and MDs) when performing a reserved procedure delegated by a MD.

CTs which had obtained, apart from their CT degree, additional degrees that allowed them to perform similar reserved procedures independently (e.g., MD or PA degrees) were excluded from the analyses. SPSS version 25 was used.

**2.5.2 Qualitative data analysis.** Focus group interviews and individual expert interviews were digitally recorded and transcribed verbatim prior to being summarized. Summaries were presented to participants through a member-check procedure [28]. Observation notes were used to make summaries and also presented through a member-check procedure. Following positive confirmation, transcripts and observation-summaries were anonymized and analyzed using an 'editing analysis style' [29], whereby QUAL data were categorized using a code-book, based on the conceptual framework [22]. Throughout the analysis phase, during dedicated team meetings, codes were added and/or specified if deemed necessary. Every 6–7 transcripts, the team evaluated whether data saturation had been reached [30]. MdH and MBB coded and analyzed the data and all team-members were involved in the interpretation of the results. NVivo version 11 was used.

## 2.6 Data triangulation

An existing triangulation protocol for combining multiple forms of QUAL data in health services research [31] was used to guide the triangulation process. Per variable, QUAN and QUAL data were sorted in relation to the research questions by means of the conceptual framework [22]. Subsequently, for each specific part the level of convergence and/or divergence between QUAN and QUAL data were categorized by two team-members. Finally, all triangulated data were discussed and interpreted together by all team-members to answer the overarching research question(s). Disagreements in convergence coding were discussed by all team-members. More details of the triangulation approach are published elsewhere [32].

## 2.7 Ethical considerations

All participants of the study were informed that their participation was voluntary and that they could, without providing a reason, cease to participate. For legal reasons, CTs recruited patients. For practical reasons, CTs also recruited MD participants because it was not known

beforehand which MDs work with CTs. Patients were informed that the study was observational (and not an intervention) and received a minimum five-days reflection period prior to given their (written) consent. Patients had the opportunity to discuss their (potential) participation with an independent expert.

The Ethics committee of the Maastricht University Medical Center (MUMC+) approved the study in August 2015 (Reference: METC 15-4-176). The study was not considered subject to the Medical Research Involving Human Subject Act (in Dutch: *Wet Medisch-wetenschappelijk onderzoek met mensen*), due to the observational set-up of the study.

## 3. Results

As previously mentioned, due to a smaller number of CTs that performed reserved procedures during all three measurement points, statistical testing for QUAN data was limited to key structure variables (e.g., sex, age). All other variables are presented through descriptive statistics (e.g., numbers, percentages). Furthermore, due to a low patient response (n = 13), only QUAL data for patients are presented here.

Results are presented by means of the framework [22], whereby i) structure variables are presented first, followed by ii) process variables, and finally iii) outcome variables. For each part, QUAN data are presented first followed by QUAL data, where available. QUAL quotations are presented in italics, whereby square brackets are used to identify where quotations have been altered ensure participant's anonymity or to improve the readability. The presented quotations are representative of the overall QUAL results. Following the QUAN and QUAL results, the triangulated results are presented.

### 3.1 Results structure variables

All graduated CTs were eligible to participate (T1 n = 215, T2 n = 284, T3 n = 315), however, due to missing contact information slightly less CTs actually received an invitation (T1 n = 189, T2 n = 271, T3 n = 302). In total, 258 CTs, 55 MDs and 13 patients participated over three QUAN measurement points. QUAL data consisted of 69 individual interviews (27 CTs, 19 MDs, 6 patients, 17 stakeholders), 3 focus group interviews (23 CTs) and 9 observations (9 CTs). In total 46 individual CTs participated in the QUAL data collection, of which 11 CTs participated in more than one data collection technique.

Table 2 presents the background characteristics for both CTs that used the ESP as well as those that did not. With the exception for 'age', the results showed no statistically significant differences over time. For T2 and T3 a statistically significant difference was identified for 'institute type' between CTs performing reserved procedures (ESP) and those that did not (no ESP). CTs that worked in 'academic hospitals' tended to perform more reserved procedures.

Responding MDs were mostly female (T1: 50.0%, T2: 64.0%, T3: 69.6%) with a mean age (year, SD) of around 42 years (T1: 43.2 (9.0), T2: 41.6 (12.3), T3: 42.9 (11.5)). The majority of MDs was employed at academic hospitals (T1: 62.5%, T2: 76.0%, T3: 60.9%) and had completed a medical specialization (T1: 87.5%, T2: 68.0%, T3: 87.0%). The medical disciplines/specializations reported most frequently were nuclear medicine (T1: 31.3%, T2: 16.0%, T3: 17.4%) and radiology (T1: 0.0%, T2: 20.0%, T3: 13.0%).

The QUAL results show that most CT graduates that used the ESP were labeled as 'PhD-student', however it also indicated that there were variations between how institutions labeled CTs. While most were in 'research positions', respondents suggested that this was mostly due to financial and organizational barriers.

**Table 2. Structure variables CTs.**

| | T1 | | | T2 | | | T3 | | | T1–3 | T1–3 | T1–3 |
|---|---|---|---|---|---|---|---|---|---|---|---|---|
| | CT ESP | | CT No ESP | CT ESP | | CT No ESP | CT ESP | | CT No ESP | Difference between measurements per group | | |
| Structure variable<br>Statistical test | N = 42<br>N (%) | | N = 67<br>N (%) | N = 52<br>N (%) | | N = 130<br>N (%) | N = 52<br>N (%) | | N = 142<br>N (%) | CT All | CT ESP | CT No ESP |
| **Sex**<br>*Female*<br>Chi$^2$ test (p-value) | 22 (52.4) | (1.00) | 36 (43.9) | 28 (53.8) | (1.00) | 70 (53.8) | 29 (55.8) | (1.00) | 81 (57.0) | (0.72) | (0.86) | (0.73) |
| **Age (in years)**<br>Mean (SD)<br>Delta<br>Independent T-test (p-value)<br>Kruskal Wallis | 28.4 (1.6) | -0.3 (0.41) | 28.1 (1.6) | 28.2 (1.9) | +0.4 (0.32) | 28.6 (2.0) | 28.9 (2.2) | 0.0 (0.97) | 28.9 (2.0) | (0.02)* | (0.34) | (0.04)* |
| **Master-specialization**<br>*Robotica & Imaging*<br>*Medical Signaling*<br>*Reconstructive Medicine*<br>*Medical Imaging & Interventions*<br>*Medical Sensing & Stimulation*<br>*Multiple answers*<br>Chi$^2$ test (p-value)<br>Fisher-exact test (p-value) | 18 (42.9)<br>14 (33.3)<br>7 (16.7)<br>1 (2.4)<br>1 (2.4)<br>1 (2.4) | N/P | NDA | 20 (38.5)<br>12 (23.1)<br>3 (5.8)<br>11 (21.2)<br>6 (11.5)<br>- | (0.31)<br>(0.31) | 43 (33.1)<br>30 (23.1)<br>15 (11.5)<br>18 (13.8)<br>16 (12.3)<br>8 (6.2) | 20 (38.5)<br>6 (11.5)<br>2 (3.8)<br>15 (28.8)<br>9 (17.3)<br>- | (0.34)<br>(0.34) | 38 (26.8)<br>27 (19.0)<br>15 (10.6)<br>33 (23.2)<br>26 (18.3)<br>3 (2.1) | N/P | N/P | N/P |
| Background variable<br>Statistical test (p-value) | N = 42<br>N (%) | | N = 67<br>N (%) | N = 52<br>N (%) | | N = 130<br>N (%) | N = 52<br>N (%) | | N = 142<br>N (%) | CT All | CT ESP | CT No ESP |
| **Type of Institution**<br>*Academic hospital*<br>*General hospital*<br>*Top clinical hospital*<br>*University*<br>*Private sector*<br>*Other*<br>*Multiple answers*<br>Chi$^2$ test (p-value)<br>Fisher-exact test (p-value) | 29 (69.0)<br>4 (9.5)<br>4 (9.5)<br>-<br>-<br>1 (2.4)<br>4 (9.5) | N/P | NDA | 38 (73.1)<br>1 (1.9)<br>7 (13.5)<br>5 (9.6)<br>-<br>1 (1.9)<br>- | <0.01**<br><0.01** | 56 (43.1)<br>2 (1.5)<br>20 (15.4)<br>19 (14.6)<br>19 (14.6)<br>14 (10.8)<br>- | 36 (69.2)<br>1 (1.9)<br>9 (17.3)<br>-<br>1 (1.9)<br>1 (1.9)<br>4 (7.7) | <0.01**<br><0.01** | 54 (38.0)<br>5 (3.5)<br>19 (13.4)<br>11 (7.7)<br>31 (21.8)<br>8 (5.6)<br>14 (9.9) | N/P | N/P | N/P |
| **Job title**<br>*Clinical Technologist*<br>*Clinical Technologist Fellow*<br>*Clinical Technologist/ PhD student*<br>*PhD student*<br>*Post-doctoral research*<br>*Other medical job title (e.g., MD)* | 3 (7.1)<br>4 (9.5)<br>4 (9.5)<br>18 (42.9)<br>2 (4.8)<br>11 (26.2) | | NDA | 7 (13.5)<br>2 (3.8)<br>12 (23.1)<br>24 (46.2)<br>-<br>7 (13.5) | | NDA | 5 (9.6)<br>4 (7.7)<br>12 (23.1)<br>18 (34.6)<br>1 (1.9)<br>12 (23.1) | | NDA | N/P | N/P | N/P |

* = p-value < 0.05

** = p-value < 0.01

N/P = Not possible to perform

NDA = No data available

> *"Currently they can appoint me as an 'extra' [research position versus clinical position]. And that is very convenient [. . .] otherwise they would have to reduce the Full Time Equivalent [of clinical staff] by 0.2 FTE. So, somebody would lose out."*

*Focus group, CT*

Labeling CTs as researchers made financing their positions easier and in addition, the label 'CT' was simply not available in most institutions. This proved problematic in certain institutions because it excluded CTs from accessing vital patient data in the electronic patient files.

*"[. . .] I have heard from [CT] colleagues that there are discussions [on ICT rights] [. . .], and that they do not [have access to files]. But frequently they circumvent this by using a different colleague's name [e.g. MD]."*

*Expert interview, CT*

To circumvent this problem, several CTs were registered as 'MDs' within their institution or could use the credentials of MDs.

*"I know for a fact that many [CTs in this institution] are registered as a 'MD-researcher."*
*Expert interview, CT*

QUAL results also identified that certain medical disciplines were more open to hiring CTs (e.g., radiology and cardiology) than others (e.g., internal medicine). Respondents suggested this was closely linked to how the medical disciplines had traditionally viewed technological change. Disciplines which had incorporated more technological elements in the past, seemed more welcoming to CTs and their ESP.

*"The average internal medicine department lags behind [other departments in terms of CTs and the use of medical technology]."*

*Expert interview, CT*

Likewise, QUAL respondents indicated that due to their combined tasks of patient care, education and research, CTs were more likely to work in academic hospitals. Another reason for this, according to many respondents, is that academic hospitals are publicly funded in the Netherlands, and thus could experiment more with emerging professional roles.

*"[Financing of CTs] in Academic Hospitals does work because everybody is on the pay-roll and the board of directors can just decide 'yes we want to experiment with the CT'. But in general hospitals, where you work in independent partnerships, it is more difficult to create such a position."*

*Expert interview, Stakeholder*

*"What also matters is finance. I mean 'Hello! Who is going to pay for a CT in a general [non-academic] hospital?.'"*

*Expert interview, Stakeholder*

While respondents expected a rise in the number of CTs in non-academic hospitals, most stakeholder groups and hospital managers expressed this would not occur overnight and that the number of CTs working in non-academic hospitals would likely remain lower than in academic hospitals.

*"[. . .] it [the employment of CTs] will also seep through to the non-academic hospitals, but to a lesser extent [. . .]."*

*Expert interview, Stakeholder*

### 3.2 Results process variables

While the overall percentages of CTs performing intake consultations between the three measurement points decreased, the mean number of intake consultations per week increased

**Table 3. Mean number of individual intake- and control consultations.**

| Type of consultation | T1 CT n = 41* n (%) mean (sd) | T2 CT n = 52 n (%) mean (sd) | T3 CT n = 52 n (%) mean (sd) | T1 MD n = 16 n (%) mean (sd) | T2 MD n = 25 n (%) mean (sd) | T3 MD n = 21 n (%) mean (sd) |
|---|---|---|---|---|---|---|
| *Intake-consultations* | | | | | | |
| Mean (SD) number of individual consultation per week | 19 (46.3) 1.8 (1.4) | 22 (42.3) 3.0 (3.5) | 19 (36.5) 3.6 (3.3) | 14 (87.5) 3.6 (3.8) | 11 (44.0) 4.3 (4.0) | 9 (42.9) 3.8 (3.0) |
| *Control-consultation* | | | | | | |
| Mean (SD) number of individual consultation per week | 15 (36.6) 1.9 (2.5) | 22 (42.3) 3.8 (5.3) | 14 (26.9) 2.6 (2.4) | 12 (75.0) 8.8 (11.3) | 10 (40.0) 9.0 (13.3) | 8 (38.1) 6.4 (7.4) |

* Not all respondents completed this part of the survey. Respondents with a mean above 70hours per week were excluded due to implausible results.

between the three measurement points. The percentage CTs performing control consultations remained similar in the first two measurements, however a drop was visible in the last measurement. Simultaneously, the mean number of control consultations performed per week fluctuated throughout the study.

For MDs, the percentage performing intake consultations was higher for the first measurement point than for the other two, while the mean number of intake consultations fluctuated between measurements points. A similar picture is seen for control consultations.

An overview of the QUAN results for intake- and control consultations is provided in Table 3.

The (total) mean time CTs spent on patient related activities increased throughout the three measurement points, and CTs spent the most (mean) time on conducting patient consults and performing medical procedures. The (total) mean time CTs spent on non-patient related activities decreased throughout the three measurement points, and CTs spent the most (mean) time on conducting scientific research.

For MDs, the (total) mean time spent on patient related activities decreased throughout the three measurement points, and MDs spent the most (mean) time on performing medical procedures. The (total) mean time MDs spent on non-patient related activities decreased throughout the three measurement points, and MDs spent the most (mean) time on conducting scientific research.

An overview of the QUAN results in relation to patient- and non-patient related activities is provided in Tables 4 and 5.

The QUAL results highlighted that the time spent on intake- and control consultations varies between individual CTs. Also, the role of CTs differed between individual CTs, institutions and (medical)disciplines. While some CTs had similar roles as MDs, others had specific roles adjusted for their expertise/background.

*"[. . .] here we have overlap [between the roles of CT and MD]. [. . .] we don't have a strict task-differentiation. We both do it [. . .]." Expert interview, CT*

*"You are also involved in a biopsy-program for patient with [type disease]. You do this once every four weeks, and you rotate [this responsibility] with CTs and MDs [researchers]." Participant observation*

The extent to which CTs were able to be involved in patient-related activities was dependent on their other responsibilities, such as (scientific)research. While at times the status of a research project allowed for more clinical hours, at other times this was not possible.

**Table 4. Mean time spent on patient related activities.**

| Activity type | T1<br>CT<br>n = 38*<br>n (%)<br>mean (sd) | T2<br>CT<br>n = 50*<br>n (%)<br>mean (sd) | T3<br>CT<br>n = 48*<br>n (%)<br>mean (sd) | T1<br>MD<br>n = 13*<br>n (%)<br>mean (sd) | T2<br>MD<br>n = 14*<br>n (%)<br>mean (sd) | T3<br>MD<br>n = 9*<br>n (%)<br>mean (sd) |
|---|---|---|---|---|---|---|
| *Patient consults*<br>Mean (SD) time spent per week | 24 (63.2)<br>3.0 (2.7) | 35 (70.0)<br>3.4 (3.4) | 35 (72.9)<br>2.9 (3.0) | 11 (84.6)<br>8.5 (8.7) | 10 (71.4)<br>7.1 (5.3) | 7 (77.8)<br>4.7 (5.0) |
| *Medical procedures*<br>Mean (SD) time spent per week | 31 (81.6)<br>5.5 (6.3) | 43 (86.0)<br>4.6 (4.3) | 41 (85.4)<br>4.9 (4.9) | 10 (76.9)<br>11.4 (10.8) | 13 (92.9)<br>12.6 (11.2) | 8 (88.9)<br>10.3 (8.2) |
| *Visitations*<br>Mean (SD) time spent per week | 7 (18.4)<br>1.7 (1.3) | 12 (24.0)<br>2.6 (2.8) | 13 (27.1)<br>2.2 (2.2) | 4 (30.8)<br>1.8 (1.7) | 7 (50.0)<br>1.4 (0.8) | 7 (77.8)<br>3.3 (3.2) |
| *Consults about individual patients*<br>Mean (SD) time spent per week | 34 (89.5)<br>1.9 (1.2) | 41 (82.0)<br>2.0 (1.4) | 42 (87.5)<br>2.8 (1.7) | 11 (84.6)<br>2.5 (1.8) | 14 (100.0)<br>5.5 (5.9) | 9 (100.0)<br>3.4 (2.4) |
| *Non-patient bound consults*<br>Mean (SD) time spent per week | 31 (81.6)<br>3.2 (2.1) | 42 (84.0)<br>3.4 (2.2) | 41 (85.4)<br>4.5 (2.6) | 10 (76.9)<br>7.2 (6.5) | 12 (85.7)<br>4.1 (2.4) | 9 (100.0)<br>4.9 (8.1) |
| *Multi-disciplinary consults*<br>Mean (SD) time spent per week | 21 (55.3)<br>2.5 (1.5) | 35 (70.0)<br>2.0 (1.5) | 42 (87.5)<br>2.8 (1.6) | 11 (84.6)<br>2.4 (1.0) | 12 (85.7)<br>3.8 (3.1) | 9 (100.0)<br>3.6 (1.3) |
| *Patient related activities (total)*<br>Mean (SD) time spent per week | 37<br>12.7 (7.2) | 50<br>12.8 (6.6) | 47<br>13.3 (8.3) | 12<br>30.8 (13.1) | 14<br>29.8 (13.5) | 9<br>27.6 (10.8) |

* Not all respondents completed this part of the survey. Respondents with a mean above 70 hours per week (for either patient or non-patient related activities) were excluded due to implausible results.

*"[when] you are at the end of your PhD project you have to finish your dissertation and there is nearly no time for clinical work." Expert interview, CT*

Overall, the QUAL results highlighted that the role of CTs within patient care and the time they could spend on it, was highly dependent on the individual as well as the setting in which they worked. CTs explained that their role, and time they could spend on patient-related

**Table 5. Mean time spent on non-patient related activities.**

| Activity type | T1<br>CT<br>n = 38*<br>n (%)<br>mean (sd) | T2<br>CT<br>n = 50*<br>n (%)<br>mean (sd) | T3<br>CT<br>n = 48*<br>n (%)<br>mean (sd) | T1<br>MD<br>n = 13*<br>n (%)<br>mean (sd) | T2<br>MD<br>n = 14*<br>n (%)<br>mean (sd) | T3<br>MD<br>n = 9*<br>n (%)<br>mean (sd) |
|---|---|---|---|---|---|---|
| *Administration*<br>Mean (SD) time spent per week | 32 (84.2)<br>3.0 (1.9) | 47 (94.0)<br>3.8 (5.6) | 48 (100.0)<br>3.3 (1.9) | 11 (84.6)<br>4.8 (5.1) | 11 (78.6)<br>4.6 (2.6) | 9 (100.0)<br>5.6 (7.7) |
| *Training/schooling*<br>Mean (SD) time spent per week | 33 (86.8)<br>1.9 (0.9) | 46 (92.0)<br>2.5 (1.5) | 46 (95.8)<br>3.2 (1.4) | 11 (84.6)<br>2.1 (0.9) | 13 (92.9)<br>2.2 (2.7) | 9 (100.0)<br>2.7 (1.3) |
| *Scientific research (e.g., Phd)*<br>Mean (SD) time spent per week | 37 (97.4)<br>24.1 (11.2) | 49 (98.0)<br>22 (11.4) | 48 (100.0)<br>22.1 (11.5) | 11 (84.6)<br>13.6 (14.7) | 13 (92.9)<br>10.3 (10.1) | 9 (100.0)<br>7.9 (9.5) |
| *Quality improvement projects*<br>Mean (SD) time spent per week | 18 (47.4)<br>2.4 (1.9) | 21 (42.0)<br>2.8 (2.6) | 40 (83.3)<br>3.0 (2.9) | 9 (69.2)<br>1.5 (1.1) | 9 (64.3)<br>1.3 (0.7) | 9 (100.0)<br>1.4 (0.5) |
| *Teaching*<br>Mean (SD) time spent per week | 26 (71.1)<br>2.5 (3.2) | 30 (60.0)<br>2.7 (2.8) | 42 (87.5)<br>2.5 (1.2) | 10 (76.9)<br>2.3 (1.3) | 13 (92.9)<br>2.6 (2.9) | 9 (100.0)<br>2.2 (1.1) |
| *Other*<br>Mean (SD) time spent per week | 11 (28.9)<br>7.0 (8.7) | 5 (10.0)<br>8.3 (9.0) | 9 (18.8)<br>10.6 (10.2) | 0 (0.0)<br>- | 2 (14.3)<br>25 (26.9) | 1 (11.1)<br>3 (-) |
| *Non-patient related activities (total)*<br>Mean (SD) time spent per week | 38 (100.0)<br>32.6 (11.1) | 50 (100.0)<br>31.1 (11.3) | 48 (100.0)<br>30.4 (9.8) | 12 (92.3)<br>25.1 (12.6) | 14 (100.0)<br>22.1 (14.9) | 9 (100.0)<br>20.1 (11.1) |

* Not all respondents completed this part of the survey. Respondents with a mean above 70hours per week (for either patient or non-patient related activities) were excluded due to implausible results.

activities, was highly influenced by the willingness of management and supervising MDs to provide those opportunities.

*"I think that [the role] is very dependent on the department, in [name hospital] that is the case, it is very different per department." Focus group, CT*

*"[. . .] first they [MD or management] have to allow or trust you to do it!" Expert interview, CT*

### 3.3 Results outcome variables

**3.3.1 Use of ESP by CTs (effectiveness).** The percentage of CTs that indicate, perform and delegate reserved procedures throughout the study decreased, while the mean number of reserved procedures performed fluctuated between measurement points. Nevertheless, the variance in the mean number of reserved procedures performed was high. For MD's the mean number of reserved procedures delegated throughout the study decreased between measurement points, however the variance was also high.

While the mean number of reserved procedures performed by CTs fluctuated throughout the study, the percentage of procedures indicated by CTs without consulting an MD increased between measurement points. Furthermore, the percentage of reserved procedures delegated to CTs after a verbal order decreased between measurement points (Fig 2).

For the reserved procedures 'defibrillation', 'elective cardioversion' and 'endoscopy', the number of respondents were too low to determine mean number of procedures per month or mean duration. No CTs performed 'lithotripsy' throughout the study. For the reserved procedures 'catheterizations', 'injections', 'ionizing radiation', 'punctures' and 'surgical procedures', the percentages of CTs performing such procedures either remained stable or increased slightly.

The QUAL results showed that 'catheterizations', 'injections', 'ionizing radiation', 'punctures' and 'surgical procedures' were done most frequently by CTs. No CTs were found that performed 'lithotripsy' during the study. Several CTs, MDs and stakeholders implored policy makers to expand the ESP since several CTs also performed another reserved procedure, namely: 'prescribing medicine'.

*"No, I do not find the composed list of reserved procedures [for the ESP of CTs] sufficient, 'prescribing medicine' is really missing from that list." Expert interview, CT*

**3.3.2 Impact of ESP on efficiency of care (efficiency).** The total number of respondents proved too small to analyze the duration per category of reserved procedure as originally planned. Instead, durations from all three measurement points were pooled per indication procedure to allow comparison between the duration of performance of reserved procedures by CTs based on their level of independence. The QUAN results show that reserved procedures indicated, performed and/or delegated by CTs independently, on average, take less time than when a CT consults an MD (or other authorized professional) (Fig 3).

Within the QUAL data, both CTs and MDs state that the ESP increased the efficiency of patient-care processes.

*"If you did not have the independent authority, supervision by a MD would be mandatory. The independent authority, in your situation, advances the efficiency of care processes." Participant—observation*

Respondents emphasize that consultations between healthcare professionals is beneficial but not always necessary.

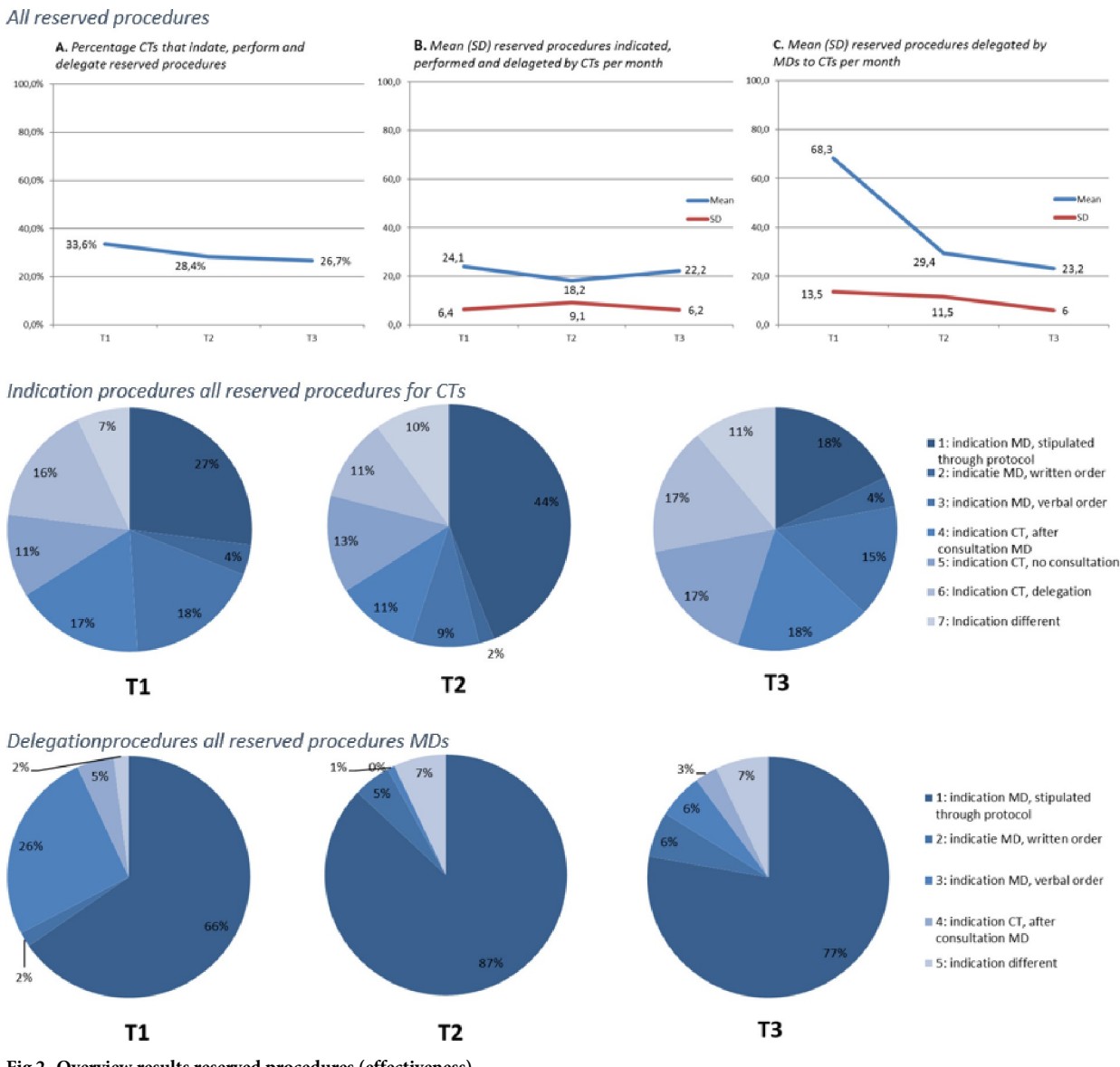

**Fig 2. Overview results reserved procedures (effectiveness).**

*"It is less efficient if you have to ask permission every time. Of course it is good to consult with others, but it is not always necessary and then it is not efficient." Expert interview, CT*

MDs indicated to be particularly pleased that CTs disturbed them less for non-essential consultation during other work activities, which also increased efficiency for those tasks. Both MDs and CTs stress that the reserved procedures performed by CTs were often not regular task shifting but were part of an innovative (often academic) research project aimed at creating new, or improving existing, medical technologies. The ESP also improved the efficiency of such research activities which pleased both MDs and CTs.

*"Well, a specific benefit [of the independent authority for CTs] is that they are very efficient for imaging problems. Consequently, we [the MDs] need only be indirectly involved. And, in*

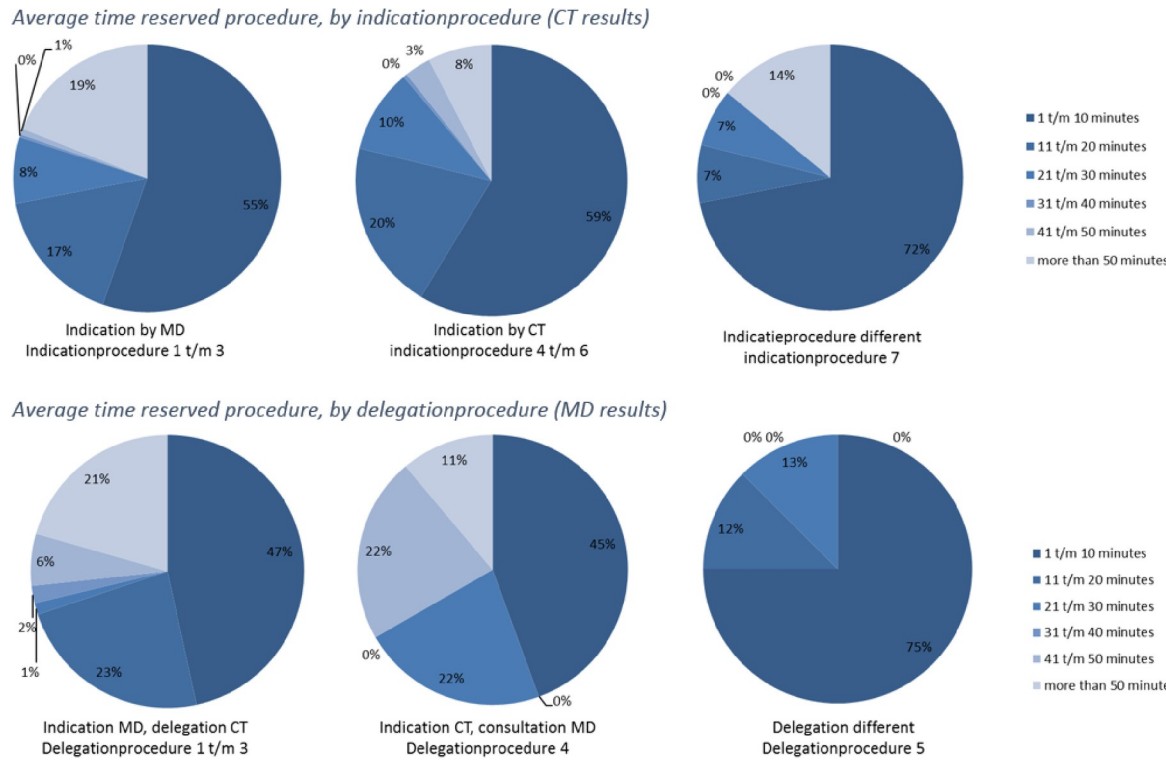

**Fig 3. Overview results reserved procedures (efficiency).**

*part, they are really doing new things, there is little 'substitution', and I find it nice that this can happen independently, I think this will eventually be good for the process of care." Expert interview, MD (and stakeholder).*

### 3.4 Triangulated data

For structure variables, both QUAN and QUAL data indicated CTs who used their ESP tend to work in academic hospitals and within technology-based medical disciplines. QUAL data suggested that this was due to the financing of academic hospitals in the Netherlands as well as their innovating character. QUAN data indicated that those CTs whom did not use the ESP were either in research jobs or working outside of (direct) patient care.

For process variables, both QUAN and QUAL data show that CTs had various roles. However, both QUAN and QUAL results suggest that the ability for CTs to spend time on patient care was dependent on other responsibilities (e.g., status of research projects) and a willingness by others (e.g., MD and management). QUAL data indicated that without such support individual CTs were unlikely to be able to fully benefit from the ESP.

For outcome variables, both QUAN and QUAL confirmed CTs use their ESP for five reserved procedures (catheterizations, surgical procedures, procedures with radiation, injections and punctures). Inconclusive QUAN data was collected for three reserved procedures (endoscopy, elective cardioversion and defibrillation) because only a handful of CTs performed such procedures and QUAL data confirmed this to be case. Neither QUAN nor QUAL results indicated CTs performed 'lithotripsy' throughout the evaluation period. QUAL data highlighted that some CTs were possibly performing another reserved procedure which was

**Table 6. Summarized results matrix.**

| Structure variables | |
|---|---|
| QUAN | QUAL |
| **CT/MD background characteristics**<br>• *No (meaningful) statistical differences* | **CT/MD background characteristics**<br>• *Most CTs using ESP work as 'phd-student/researchers'* |
| **Patient background characteristics**<br>• *Not presented due to low response* | **Patient background characteristics**<br>• *Patient background varied in age, ethnicity and diagnosis* |
| **Organizational features**<br>• *CTs working in Academic Hospitals use ESP more than other organizations* | **Organizational features**<br>• *Organization label CTs differently due to financial and organizational reasons*<br>• *Academic hospitals and 'technology' based disciplines more prone to hire CTs* |
| **Process variables** | |
| QUAN* | QUAL |
| **Intake and control consultations (CTs/MDs)**<br>• *CTs performing (and mean number of) intake- and control vary* | **Intake and control consultations (CTs)**<br>• *Time spent on intake- and control consultations varies between CTs* |
| **Weekly activities (CTs/MDs)**<br>• *For CTs total time spent on patient tasks increased* | **Weekly activities (CTs)**<br>• *CT role different between individual CTs, institutions and discipline*<br>• *Extent to which CT was involved in patient care very dependent on attitude supervisor and setting* |
| **Outcome variables** | |
| QUAN* | QUAL |
| **Effectiveness**<br>• *Percentage CTs using ESP decrease, average number of procedures fluctuate*<br>• *Percentage of procedures performed without consulting MD increased*<br>• *Average number of procedures delegated from MD to CT decreased*<br>• *CTs performed 5 procedures often, 3 sporadically and 1 never* | **Effectiveness**<br>• *CTs use their ESP to perform 5 reserved procedures often, 3 sporadically and 1 never*<br>• *Respondents called to expand ESP, since several CTs also perform reserved procedure: 'prescribing medicine'*<br>• *CTs limited in performing reserved procedures due to structure and process variables.* |
| **Efficiency**<br>• *Procedures indicated, performed and/or delegated by CTs alone take less time* | **Efficiency**<br>• *CTs and MDs state ESP increases the efficiency of patient care and research processes.*<br>• *MDs pleased with ESP for CTs*<br>• *ESP for CT is not regular task shifting* |

\* *Only descriptive statistics presented*

outside the scope of the study, namely: 'prescribing of medicine'. Both the QUAN and QUAL data suggest that an ESP increased the efficiency of care provided by CTs.

A summary of the triangulated data is presented in a Results Matrix (see Table 6).

## 4. Discussion

This study aimed to evaluate whether a legal amendment providing a temporary independent authority to indicate, perform and/or delegate nine categories of reserved procedures (ESP) to CTs would result in CTs performing more, and MDs performing less, reserved procedures independently and that the duration of the procedures performed by CTs would decrease over time. The combined mixed methods results of the study highlight that CTs use their ESP frequently to perform five categories of reserved procedures independently (i.e., catheterizations, surgical procedures, procedures with radiation, injections and punctures). As a result, the ESP seems effective for these five reserved procedures. Both QUAN and QUAL data confirm that

only a limited number of CTs use their ESP for performing three other categories of reserved procedures (i.e., endoscopy, elective cardioversion and defibrillation). Both QUAN and QUAL data show no CTs using their ESP to perform procedures for one category of reserved procedures (i.e., lithotripsy). Both QUAN and QUAL data suggest that the ESP most likely increased the efficiency of care delivery. Based on the results of the study, it was recommended to provide a permanent independent authority for CTs to perform the categories of five reserved procedures: catheterizations, surgical procedures, procedures with radiation, injections and punctures. Additional research was recommended to investigate the impact of the ESP for three reserved procedures (i.e., endoscopy, defibrillation and elective cardioversion). Lastly, it was recommended to discontinue the ESP for one reserved procedure (i.e., lithotripsy) immediately since no CTs performed this procedure, and therefore was found neither effective nor efficient. In 2019 the Minister of Health announced to take over these recommendations and provide CTs with a permanent independent authority to perform five reserved procedures. Dutch parliament voted on an amendment of the ICHP-Act in April 2020 and in July 2020 the CT was officially provided a permanent legal authority to independently perform, indicate and delegate five reserved procedures [33].

In addition, this case study offers some tentative lessons regarding shifting tasks to emerging professional groups. Firstly, the results seem to indicate that the use of ESP is dependent on the setting in which CTs work. Various barriers for the introduction of emerging professional groups (e.g., no or limited access to patient files and no structural financing), limited the deployment of CTs which likely affected the use of ESP by CTs during the evaluation period. It can be assumed that acceptance of an emerging professional group by established groups is not self-evident and necessitates concrete action to stimulate, or at least, limit administrative and legal barriers. Such challenges are associated with the implementation of an ESP in more general. A detailed overview of implementation barriers and their potential influence on ESP is presented elsewhere [34]. These findings are in line with previous studies, which have highlighted that established professional groups (e.g. NP/PA's) face similar implementation challenges [2, 35, 36].

Secondly, the results indicate that CTs frequently had additional tasks (e.g., conducting scientific research) which limited time for providing patient care and thus restricted their use of the ESP. This could potentially explain why the number of CTs that used the ESP during all three measurement points was lower than expected, and as such, was likely more a reflection of the current role that CTs have in Dutch healthcare rather than a measure for determining the impact of the legal amendment. Also, it can be expected that CTs have to grow into their role as healthcare provider, making it likely that the ESP will be used more often in the future. This is in line with the development of other Allied Healthcare Professional groups such as PA's, as reported in previous studies [37, 38]. As such, the current study indicates that it is important not to compare the tasks of an emerging professional group with those of an established professional group right away, since roles might differ substantially and/or might take time to be established properly.

Third, the findings of the current study confirm results of previous studies on task shifting between established professional groups [2, 35, 36, 39] and suggest that task shifting remains, to a great extent, dependent on the willingness by physicians to take part in it. While research into the attitude of Dutch physicians in relation to reserved procedures has been done before [40], a comparison to the present situation is of limited relevance because it pre-dates the introduction providing professional groups an ESP. Nevertheless, while research has been done on task shifting between established professional groups [40, 41], we are unaware of studies focusing on task shifting to emerging professional roles. Consequently, additional research should be done to compare physicians' attitudes to task shifting to emerging professional roles

and established professional groups. This may provide policymakers with additional insights into how they can overcome such difficulties.

Fourthly, while task shifting refers to the transfer of work and responsibilities from one professional group to another [1], tasks are usually shifting vertically (from higher cadre professionals to lower cadre professionals). However, this is not the case for CTs since they, like physicians in the Netherlands, have at least six years university schooling (similar to obtaining a MD degree in the Netherlands). Furthermore, many CTs continue their education in the form of a PhD or fellowship to specialize in a particular field. As a result, CTs are merely educated differently than MDs. Interestingly, we are unaware of any studies which explored horizontal redistribution of tasks to new professional groups, although attention has been given to shifting tasks horizontally between medical specialists [42]. From studies focusing on horizontal task shifting between established professional groups it shows that this takes place predominantly in more developed economies to either decrease patient referral, improve overall (cost-) efficiency [43], or in response to the development of new technology [9]. Examples include shifting tasks from medical specialists in a hospital setting to General Practitioners [44], or shifting tasks from cardiothoracic surgeons to interventional surgeons due to developments in technology [45]. In this study not only was task shifting from physicians to CTs horizontal, it also involved an emerging professional group, which likely led to additional implementation problems. Emerging professional groups are by design 'groundbreakers', and the CT profession was no exception. They emerged as a professional group to fill the gap between "engineer" and "physician" [11] and an important part of their role is to "design and develop new diagnostic and therapeutic opportunities" [46]. While the results of this study indicate that the legal amendment to provide an ESP to CTs was likely effective and efficient for a small portion of CT graduates, a larger portion of CT graduates did not use the ESP at all during the evaluation period. A possible explanation for that lies in the findings of structure and process variables within our study, which highlight the difficulties for newly graduated CTs to be positioned correctly within patient care. Instead of directly providing patient care, many CTs were forced to first pursue a research position with limited access to patients and financial security, often within academic hospitals due to financing structures of Dutch Healthcare. It is likely such barriers had an enormous impact on the use of the temporary ESP. The most important lesson therefore is that policymakers should be more aware of systemic factors which might influence task shifting, such as organization, financing and skepticism from existing professional groups.

Finally, while several studies have begun to explore responsibility issues in relation to new technologies [47, 48], such studies tend to focus on the responsibilities between physicians and the manufacturer of a technology and not on the responsibilities of tasks between different professional groups in healthcare. The increasing uptake of complex and self-learning technology in healthcare, gives rise to the question whether MDs interaction with new technology is redefining their role and position. Following from this, we encourage more research on task shifting to emerging professional groups and between professional groups with similar educational levels, and with a particular focus on new healthcare technologies.

There are some limitations in this study. Firstly, a pre-posttest design was not possible because the temporary ESP had already been formalized before the study commenced. As a result, the study is reliant on retrospective QUAL data to provide insight into the pre-test situation. Secondly, because less CTs performed reserved procedures during all three measurement points, elaborate (longitudinal) statistical analysis was not possible. Consequently, with the exception of key structure variables, the QUAN analysis was limited to descriptive statistics. While a lower QUAN response limited our ability to statistically test this hypothesis and should therefore be approached with caution, the QUAL response was high and seemed to

confirm the QUAN results. Thirdly, the response rate under patients (n = 13) was too low to include QUAN data and consequently only QUAL data were included in the analysis. While the QUAL data from patients provides additional information, we should be careful not to overstate their importance. As such, the QUAL data from patients should be seen as an exploratory addition to the CT, MD and stakeholder data. Fourthly, while the ICHP-Act names the categories of reserved procedures, a definitive list of which procedures fall under each category is not specified by the IHCP-Act, presenting a practical problem for the research team. The list as used during this study was compiled on the basis of earlier research into reserved procedures in the Netherlands [21, 40], a survey amongst CTs prior to the start of the data collection and advice from medical specialists. Despite such measures, there is a possibility that the list was incomplete and therefore could contribute to an underrepresentation of the number of CTs performing reserved procedures. Finally, the terms 'effectiveness' and 'efficiency' as applied by the commissioning party, have at times caused misunderstandings. Frequently respondents or other stakeholders were under the impression the study focused on the effectiveness and efficiency of CTs in Dutch healthcare, however, in reality the study was formed to investigate whether the legal amendment to provide CTs with an ESP was used (effective) and if this saved time (efficient). Within our study the terms 'effectiveness' and 'efficiency' should not be confused with a measure process success or patient measures such as recovery time, medical errors and/or survival rates. Such misunderstandings were addressed by the research team, by including additional background information for respondents and stakeholders. Notwithstanding these limitations, this study provided rich QUAN and QUAL data because of its broad scope and study design. Also, the findings are derived from three measurement points collected over a period of 27 months. Finally, the study meets both the COREQ [49] and STROBE [50] guidelines for, respectively, conducing quantitative and qualitative research. Nevertheless, generalizations need to be made with caution since the response is low and data might be context specific and could therefore only be applicable to the role of CTs in the Netherlands.

## 5. Conclusion

This study, commissioned by the Dutch Ministry of HWS, evaluates task shifting to CTs in the Netherlands through a temporary measure to provide CTs with an independent authority for nine categories of reserved procedures. The results suggest the temporary independent authority was used (effective) and saved time (efficient) for five categories of reserved procedures. On the basis of this study, three recommendations were made to the Dutch Minister of HWS. First, provide a permanent independent authority to CTs to indicate, perform and delegate five categories of reserved procedures (catheterizations, surgical procedures, procedures with radiation, injections and punctures). Second, conduct additional research for three categories of reserved procedures (endoscopy, elective cardioversion and defibrillation). Third, discontinue the ESP for one category of reserved procedures (lithotripsy). The Minister followed the study's recommendations and provided a permanent ESP for five reserved procedures to CTs, taking effect in July 2020 [51].

While the study presents findings to shifting tasks to a specific healthcare professional in the Netherlands, it also provides tentative lessons about the challenges associated with shifting tasks to emerging professional groups in general. The results establish that providing an ESP to facilitate task shifting to such emerging professional groups will be used and can make care more efficient. The study also reveals various challenges associated with task shifting. While established professional groups can face similar challenges, it is likely that the challenges associated with task shifting are exacerbated for emerging professional groups. When pursuing

task shifting to emerging professional groups, policymakers must be aware of such differences and judge accordingly. Concretely this means incorporating systemic factors which might influence task shifting, such as organization, financing and skepticism from existing professional groups. Finally, since this study represents the first evaluation of task shifting to an emerging professional group, it is vital more research will be performed to provide additional insight into the unique challenges this poses for policymakers, healthcare professionals and other stakeholders.

## Author Contributions

**Conceptualization:** Maarten de Haan, Yvonne van Eijk-Hustings, Hubertus Vrijhoef.

**Data curation:** Maarten de Haan, Monique Bessems-Beks.

**Formal analysis:** Maarten de Haan, Yvonne van Eijk-Hustings, Monique Bessems-Beks, Daisy De Bruijn-Geraets.

**Funding acquisition:** Yvonne van Eijk-Hustings, Hubertus Vrijhoef.

**Investigation:** Maarten de Haan, Yvonne van Eijk-Hustings.

**Methodology:** Maarten de Haan, Yvonne van Eijk-Hustings, Hubertus Vrijhoef.

**Project administration:** Maarten de Haan, Monique Bessems-Beks, Hubertus Vrijhoef.

**Resources:** Maarten de Haan.

**Software:** Maarten de Haan.

**Supervision:** Maarten de Haan, Yvonne van Eijk-Hustings, Carmen Dirksen, Hubertus Vrijhoef.

**Validation:** Maarten de Haan, Hubertus Vrijhoef.

**Visualization:** Maarten de Haan.

**Writing – original draft:** Maarten de Haan.

**Writing – review & editing:** Maarten de Haan, Yvonne van Eijk-Hustings, Monique Bessems-Beks, Daisy De Bruijn-Geraets, Carmen Dirksen, Hubertus Vrijhoef.

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
