## [Decision Letter · Decision Letter 0]

8 Apr 2022

PONE-D-21-40694The Effects of Task Shifting to a New Profession in Dutch Healthcare  - A Mixed Methods Study Involving the Clinical TechnologistPLOS ONE

Dear Dr. de Haan,

Thank you for submitting your manuscript to PLOS ONE. After careful consideration, we feel that it has merit but does not fully meet PLOS ONE’s publication criteria as it currently stands. Therefore, we invite you to submit a revised version of the manuscript that addresses the points raised during the review process.

We look forward to receiving your revised manuscript.

Kind regards,

Conor Gilligan

Academic Editor

PLOS ONE

Journal Requirements:

Additional Editor Comments (if provided):

The reviewers have provided some very useful feedback and suggestions on the manuscript. Please give careful consideration to the comments and consider making changes to your paper to address this.

Reviewers' comments:

Reviewer's Responses to Questions

**Comments to the Author**

1. Is the manuscript technically sound, and do the data support the conclusions?

Reviewer #1: Partly

Reviewer #2: Yes

2. Has the statistical analysis been performed appropriately and rigorously? 

Reviewer #1: Yes

Reviewer #2: Yes

3. Have the authors made all data underlying the findings in their manuscript fully available?

Reviewer #1: Yes

Reviewer #2: Yes

4. Is the manuscript presented in an intelligible fashion and written in standard English?

Reviewer #1: Yes

Reviewer #2: Yes

5. Review Comments to the Author

Reviewer #1: The study is interesting and of relevance in its field. However, it faces several challenges (mostly related to sample size) which make it difficult if not impossible to sufficiently answer the question on effectiveness and efficiency of CT. Using the QUAL element provides additional information which is good, but the QUAN element really faces major limitations. One way would be to state from the beginning that this is an exploratory study, move away from stating that the study analyses "effectiveness and efficiency" (perhaps towards "frequency in performance of reserved medical procedures, or similar) which it cannot do statistically and tone down the conclusions in the discussion section.

Please find more detailed comments below:

Introduction

- The education of the Clinical Technologist (CT) is briefly described, which is good but this is insufficient in the level of detail: there is no mention if and how many practical trainings the CTs have as part of the studies, what courses which prepare them for the skills needed to undertake reserved procedures they have to undertake, etc.

o Later in the manuscript, it is mentioned that CTs were listed as PhD students/researchers and were doing research. Is this part of the CT education and if yes, this should be better explained in the introduction. As this is a new role, it should be better described

- “Studies evaluating task shifting to established professionals groups, like Nurse Practitioners (NP’s) and Physician Assistants (PA’s) have highlighted that it improves the quality of care, increases patient satisfaction, reduces workforce shortages as well as increases the overall effectiveness and efficiency of care (2, 4-9)”: This sentence is too generic and does not fully capture the evidence base on the effects of NPs and PAs internationally. The evidence is not as straightforward as it suggests in the sentence.

Methods

- The methods section faces several limitations: reporting should be improved: e.g. sample size calculation, recruitment, quality assurance during data collection, planned testing of the hypotheses? What were the hypotheses?

- Ethics: the name of the ethics committee was left blank as was the reference number? “The Ethics committee of the [NAME INSTITUTION] approved the study in August 2015 [REFERENCE NUMBER ETHICS COMMITTEE]. T”

- In red colour throughout the manuscript are several sources named “anonymous…”, It is unclear why in red and second, why there are so many sources with anonymous authorship.

Results:

- 13 patients? This raises major concerns in terms of generalizability (not even talking about representativeness), actually, the authors should consider omitting the findings from these 13 patients

- Sample size of the CTs is also low when it comes to the analysis of the reserved procedures. This is a major limitation. In fact, the paper should move away from stating that it analyses the “effectiveness” which implies quality of care elements as well, toward e.g. the performance and frequency of performance of a set of reserved procedures.

Discussion

- This sentence has major issues: “While a lower QUAN response limited our ability to statistically test this hypothesis, the combined results of this mixed methods study highlight that CTs use their ESP frequently to perform five categories of reserved procedures independently (i.e., catheterizations, surgical procedures, procedures with radiation, injections and punctures). As a result, the ESP seems effective for these five reserved procedures.” - it is a limitation that the sample size was not reached and should be stated as such instead of saying that together with the QUAL element, the 5 procedures seemed to be effective.

Reviewer #2: In this manuscript, the authors evaluate how shifting reserved procedures from physicians to an emerging profession, in this case clinical technologists (CTs), affects the effectiveness and efficiency of patient care. The authors use the conceptual framework of Donabedian to guide their analysis of the data gathered. Both quantitative and qualitative data were gathered resulting in multiple perspectives on the effectiveness and efficiency of task shifting.

The methods are clearly and elaborately described. The results are presented in a structured manner, following the conceptual framework used, and tables and figures are used adequately to support the interpretation of the data, although figures 2 and 3 are somewhat redundant given the elaborate presentation of findings in the main text. The conclusions are supported by the results.

\\My main concern is the readability of the manuscript. The design of the study is quite complex and many data sources are used, both quantitative and qualitative. Data from different sources are reported separately and in combination. All this makes it hard to follow which results are relevant and how they contribute to answering the overall research question. Although tables 1 and 6 are added to provide an overview of the data sources and a summary of the results, the main text has so many details that it is hard to keep track of the main outcomes of this study. This could be improved.

On page 4 the research question is introduced which focuses on evaluating effectiveness and efficiency. However, in the introduction, these terms are not conceptualized. From the hypothesis, it becomes clear that effectiveness is associated with an increase in reserved procedures performed by CTs and efficiency with duration but the definitions are not introduced or explained. I would encourage the authors to not only explain task shifting in the introduction but also the definition and measurement of effectiveness and efficiency in this particular context.

In the conclusions on pages 26 and 27, there is a strong focus on the recommendation that is made to the Dutch Ministry of HWS. However, I find the results in light of 1) horizontal task shifting and 2) the task shifting to an emerging profession much more interesting (see page 27). For example, the authors state that ‘when pursuing task shifting to emerging professional groups, policymakers must be aware of such differences and judge accordingly’. However, it is unclear to me what the authors mean by that. How can one judge accordingly, given the results of the current study? Also, to what extent are efficiency and effectiveness adequate indicators of successful task shifting from physicians to CTs? I would like to read about the authors’ perspective on that in the discussion.

6. PLOS authors have the option to publish the peer review history of their article (what does this mean?). If published, this will include your full peer review and any attached files.

Reviewer #1: No

Reviewer #2: **Yes: **Marleen Groenier

---

## [Author Response · Author response to Decision Letter 0]

8 Oct 2022

We have added a 'response to reviewers' file within this resubmission which includes a point-by-point response to all reviewer and editor comments.

---

## [Decision Letter · Decision Letter 1]

18 Nov 2022

PONE-D-21-40694R1Evaluating Task Shifting to the Clinical Technologist in Dutch Healthcare: A Mixed Methods StudyPLOS ONE

Dear Dr. de Haan,

Thank you for submitting your manuscript to PLOS ONE. After careful consideration, we feel that it has merit but does not fully meet PLOS ONE’s publication criteria as it currently stands. Therefore, we invite you to submit a revised version of the manuscript that addresses the points raised during the review process.

We look forward to receiving your revised manuscript.

Kind regards,

Conor Gilligan

Academic Editor

PLOS ONE

Journal Requirements:

Additional Editor Comments:

Thank you for your revision. The original reviewers and I have all now reviewed the revised version and will be happy to accept the paper with some very minor additional changes. Please note the comments made by both reviewers.

Some grammatical errors to correct include: on page 3, changing limited to 'limiting' in reference to the generalisability of findings from other studies; page 11 final paragraph, change 'date' to 'data'

One reviewer comments on an ongoing concern with the use of the term 'effectiveness'. I can see that you define effectiveness for your study purpose but perhaps you could provide some further clarification in the discussion to highlight the limitation of the measures of effectiveness used (no ,easure of procedure success or patient measures such as recovery time, errors etc).

Is the 258 CTs that participated the entire pool that was invited/eligible? Clarity around recruitment processes and rates is lacking.

Please ensure that the example quotes provided match the description in the text. There are some places where several points are made in a paragraph but the quote following it does not directly relate to the last point made.

Reviewers' comments:

Reviewer's Responses to Questions

**Comments to the Author**

1. If the authors have adequately addressed your comments raised in a previous round of review and you feel that this manuscript is now acceptable for publication, you may indicate that here to bypass the “Comments to the Author” section, enter your conflict of interest statement in the “Confidential to Editor” section, and submit your "Accept" recommendation.

Reviewer #1: All comments have been addressed

Reviewer #2: (No Response)

2. Is the manuscript technically sound, and do the data support the conclusions?

Reviewer #1: Yes

Reviewer #2: Yes

3. Has the statistical analysis been performed appropriately and rigorously? 

Reviewer #1: Yes

Reviewer #2: Yes

4. Have the authors made all data underlying the findings in their manuscript fully available?

Reviewer #1: Yes

Reviewer #2: Yes

5. Is the manuscript presented in an intelligible fashion and written in standard English?

Reviewer #1: Yes

Reviewer #2: Yes

6. Review Comments to the Author

Reviewer #1: The comments were addressed sufficiently and the paper has considerably improved. One issue remains: the authors still want to use the term effectiveness which is misleading in this context due to various reasons on which I elaborated earlier (e.g. sample size, study design, etc.). I would strongly recommend to replace the word with uptake or use it as rarely as possible throughout the manuscript.

Reviewer #2: The authors adequately addressed my concerns.

I urge the authors to do a final spell- and grammar check before submitting their final version. There is still quite a number of minor errors (e.g., singular/plural errors), especially in the added paragraphs in the introduction.

7. PLOS authors have the option to publish the peer review history of their article (what does this mean?). If published, this will include your full peer review and any attached files.

Reviewer #1: No

Reviewer #2: No

---

## [Author Response · Author response to Decision Letter 1]

28 Dec 2022

A point-by-point response to the comments made by the editorial team and reviewers is provided in the uploaded file "response to reviewers".

---

## [Editor Report · Decision Letter 2]

16 Jan 2023

Evaluating Task Shifting to the Clinical Technologist in Dutch Healthcare: A Mixed Methods Study

PONE-D-21-40694R2

Dear Dr. de Haan,

We’re pleased to inform you that your manuscript has been judged scientifically suitable for publication and will be formally accepted for publication once it meets all outstanding technical requirements.

Kind regards,

Conor Gilligan

Academic Editor

PLOS ONE

---

## [Editor Report · Acceptance letter]

23 Jan 2023

PONE-D-21-40694R2 

Evaluating Task Shifting to the Clinical Technologist in Dutch Healthcare: A Mixed Methods Study 

Dear Dr. de Haan:

I'm pleased to inform you that your manuscript has been deemed suitable for publication in PLOS ONE. Congratulations! Your manuscript is now with our production department. 

Kind regards, 

on behalf of

Dr. Conor Gilligan 

Academic Editor

PLOS ONE